# Temperature-Dependent Growth and Evolution of Silicene on Au Ultrathin Films—LEEM and LEED Studies

**DOI:** 10.3390/ma15041610

**Published:** 2022-02-21

**Authors:** Tomasz Jaroch, Ryszard Zdyb

**Affiliations:** Institute of Physics, Faculty of Mathematics, Physics and Computer Science, Maria Curie-Sklodowska University, Pl. M. Curie-Sklodowskiej 1, 20-031 Lublin, Poland; ryszard.zdyb@umcs.pl

**Keywords:** silicene, surface segregation, heterostructure, LEEM, LEED

## Abstract

The formation and evolution of silicene on ultrathin Au films have been investigated with low energy electron microscopy and diffraction. Careful control of the annealing rate and temperature of Au films epitaxially grown on the Si(111) surface allows for the preparation of a large scale, of the order of cm^2^, silicene sheets. Depending on the final temperature, three stages of silicene evolution can be distinguished: (i) the growth of the low buckled phase, (ii) the formation of a layered heterostructure of the low buckled and planar phases of silicene and (iii) the gradual destruction of the silicene. Each stage is characterized by its unique surface morphology and characteristic diffraction patterns. The present study gives an overview of structures formed on the surface of ultrathin Au films and morphology changes between room temperature and the temperature at which the formation of Au droplets on the Si(111) surface occurs.

## 1. Introduction

Much of the modern research related to condensed matter concerns ultrathin layered systems, which possess unique and unexpected, but often very desired, properties unknown for bulk materials. Tailor-made two-dimensional (2D) materials constitute a particularly interesting group of systems that can be used as building blocks in nanotechnology. A specific section of the 2D’s consists of a rapidly expanding family of materials composed of only one element of group 13–16 of the periodic table. Undoubtedly the most famous of them is graphene, obtained in 2004 [1], although theoretical predictions regarding monolayers of graphite appeared a long time ago [2]. The discovery of graphene sparked a lot of excitement and inspired the world of science to search for similar artificial structures made of other elements organized in a structure of honeycomb lattices. Among the others, we know of (P, Ge, Sn, B, As, Sb, Bi, Te, Se, Ga, Pb)-based Xenes—a unique subclass of artificial 2D materials [3,4,5,6,7,8,9,10,11,12,13,14,15,16,17,18,19,20,21,22,23].

This article is related to the next mono-elemental 2D material based on Si atom silicene. Due to the fact that silicene is chemically compatible with modern electronics based on silicon, it is hoped to be used as a basic functional material in miniaturized electronics and optoelectronics applications. Silicene forms two phases, namely low buckled and planar, on the atomic scale. Thus, it seems to be an exceptionally interesting material with *sp^2^*-*sp^3^* and pure *sp^2^* hybridization, respectively. Silicene, theoretically predicted in 1994 [24], has been described in numerous articles concerning its synthesis, modification and possible applications [25,26,27,28,29,30,31,32,33,34,35,36,37,38]. Like other Xenes, silicene possesses a linear band structure with Dirac cones. The cross of the π-π* band, in turn, determines its desired physical properties. According to theoretical studies, it is possible to transform silicene into a magnetic form and use this material in spintronics applications as a spin filter or spin transistor [39,40,41]. Thanks to spin-orbit coupling and, in consequence, the relatively easy tunable bandgap, not feasible for graphene, silicene may also be considered as a valley filter in valleytronics [42,43,44]. One of the silicene drawbacks is the lack of possibility to be exfoliated. It was synthesized by the Molecular Beam Epitaxy (MBE) method on various types of substrates, mainly metallic. The multi-domain character of silicene and the variety of possible superstructures depend on the substrate and synthesis conditions, in particular, the temperature and deposition rate. The dedicated substrate for the epitaxial growth of multilayer silicene is Ag(111) [45,46,47,48]. Such a substrate is commonly used for the synthesis of other Xenes. Silicene was also obtained on Ir(111) [49], ZrC(111) [50], Ru(0001) [51], Pb(111) [52,53], Si(111)√3 × √3-Ag [54,55,56]. Spontaneous surface segregation of Si atoms was used as a method to fabricate self-ordered silicene on ZrB_2_(0001) [57] and also recently on Au(111) thin films, where planar epitaxial silicene with pure *sp^2^* hybridization of Si atoms was also found [58].

The planar and low buckled phases of silicene forming layered heterostructures have been described in detail [59]. Ordered, mutually twisted layers of both silicene phases were successfully prepared by the self-organization process, which was realized by spontaneous surface segregation of Si atoms on the stack of a thermally-treated Au multilayer system. 

Here, we report on the details of the formation of silicene on ultrathin layers of Au(111) deposited on a Si(111) substrate. We carefully analyze the subsequent stages of the silicene formation by self-ordering: (i) preordering of the low buckled phase, (ii) its coexistence with the planar allotrope and (iii) final gradual decomposition of the silicene lattice. It is worth noticing that the individual stages of silicene evolution and their fingerprints can be supervised by the exact temperature control of the system. 

## 2. Materials and Methods

All experiments have been performed in a Low Energy Electron Microscope (LEEM) (Elmitec, Clausthal-Zellerfeld, Germany) under an ultra-high vacuum with a base pressure below 3 × 10^−11^ mbar. Si(111) monocrystalline substrate with a specific resistivity of 3–8 Ω cm at room temperature (RT) was used for the preparation of silicene. Figure 1 presents the Low Energy Electron Diffraction (LEED) pattern of the clean Si(111)-(7 × 7) surface (a) obtained after several flashes up to about 1500 K and the corresponding LEEM image revealing (1 × 1) and (7 × 7) domains (d). The deposition of 0.5 monolayers (ML) or 1.2 ML Au (1 ML corresponds to the density of atoms in the (111) layer of bulk Au) with subsequent annealing at 950 K and 900 K, respectively, results in the formation of superstructures: (5 × 2) with coexisting (√3 × √3) in the former and (6 × 6) in the latter case. Figure 1 presents LEED patterns (b, c) and LEEM images (e, f) of the prepared surfaces.

The Si(111)-(6 × 6)Au surface is known to promote more flat growth of Au films at RT compared to the Si substrates with other reconstructions [60]. Therefore, most of the experiments have been performed with the Si(111) surface terminated with the (6 × 6)Au superstructure. The small black spots visible in Figure 1f come from an excess of Au during the preparation of (6 × 6)Au, which forms droplets. In the next step, Au films were grown at RT. Au was deposited from the resistively heated crucible with a rate of 0.2 ML/min and pressure below 3 × 10^−10^ mbar by the molecular beam epitaxy method. In order to determine the evaporation rate, we grew Au on a W(110) substrate. The elevated temperature growth assures a monolayer-by-monolayer growth mode of the two first layers. The completion and appearance of the first and second Au layers are visible as a strong contrast change in LEEM. After the growth of the Au film (8 ML, 10 ML and 16 ML), gentle annealing was applied with an initial rate of 10 K/min, which was reduced down to 3 K/min at 370 K and 1 K/min while approaching 470 K. Such a heating rate is an order of magnitude smaller than that described in [61,62]. The presented results were obtained for 16 ML and 8 ML (the last figure) thick Au. The distance between the (00) and (1 × 1) diffraction spots of the Si(111)-(7 × 7) surface was used as a scale for the calculation of the lattice constants of the newly appeared structures on the surface. This distance corresponds to $a_{\left[11\bar{2}\right]}^{*}=1.89$ Å^−1^.

## 3. Results and Discussion

It is well known that in the considered system of Au films on Si, a significant diffusion of the Si atoms from the substrate through the gold layer to the surface of the sample occurs and increases with rising temperature. Besides the growing rate of diffusion of the Si atoms, the temperature increase also causes crystallographic ordering of the surface layer. Under a proper control of experimental conditions, the planar phase coexisting with the low buckled phase of silicene could be obtained [58,59].

It appears that depending on the annealing temperature of the Au ultrathin films, three stages in the formation and appearance of silicene could be distinguished: (i) a low buckled phase, (ii) a layered heterostructure of the low buckled and planar phases and (iii) a silicene degradation stage. In terms of the complexity of silicene degradation, the last stage may be additionally divided into two steps denoted as (1) domain multiplication and (2) domain degradation. 

### 3.1. Thermally Supported Development of the Low Buckled Phase of Silicene

The freshly prepared non-annealed RT system of rough Au film revealed a very weak order of the surface. Although the LEEM images did not show any extra features, apart from the occasionally visible Au droplets, there was a very blurry set of barely visible spots forming a ring in the LEED pattern, Figure 2a.

The existence of the diffraction spots showed that the surface layer formed in situ by the spontaneous segregation of Si atoms is not completely amorphous; instead, it reveals a very short-range order. The diffraction spots initially forming the blurry ring separated and became relatively sharp and intense upon annealing. The pink arrow in Figure 2b indicates the ring with diffraction spots of the low buckled phase of silicene (also marked by pink loops related to selected domains in Figure 3). There were 24 such diffraction spots forming four hexagons and denoting four separated domains of the silicene. The pink hexagon in Figure 2c connects spots of one of the domains. From the distance between those spots and the (00) spot, the lattice constant of 3.85 ± 0.05 Å was obtained. At this point, it is worth emphasizing that the low buckled phase of silicene was the only phase existing on the surface of the Au ultra-thin film up to about 420 K.

### 3.2. Thermally Supported Development of Layered Heterostructure of the Low Buckled and Planar Phase of Silicene

Further increase of the sample temperature by just a few degrees revealed a set of 24 new diffraction spots, which are in a bit smaller distance to the (00) one. The new pattern is a fingerprint of two coexisting phases of silicene—the low buckled one and the new planar allotrope. They are marked by pink loops and green circles, respectively, in Figure 3a.

The domains of the planar silicene grow with increasing temperature and finally cover the entire surface after annealing at the temperature range of 470–490 K. The planar phase of silicene formed four major well-developed domains visible as four sets of hexagonally arranged diffraction spots in LEED patterns, Figure 3b, Ref. [59]. Similarly, like in the STM experiments [58], the LEEM images of the planar phase reveal characteristic well-resolved hexagonal-like features, Figure 3c.

It is worth emphasizing that there was a relatively narrow range of temperatures where the structure of the planar silicene with only 24 perfectly separated and intense diffraction spots related to this phase. Besides these strongest diffraction spots (a ring indicated by the green arrow in Figure 4a), there were spots, although considerably weaker ones (the pink arrow), belonging to the low buckled silicene. This means that both phases coexist. Simultaneously, as a result of the increasing order of the surface, a number of diffraction spots related to the (√3 × √3), (√7 × √7) and (√21 × √21) superstructures appeared (the yellow, blue and black arrow, respectively). The observed additional periodicities resulted from a superposition of the low buckled and planar phases of silicene, Figure 4a [59]. According to our earlier studies, the low buckled phase of silicene is located below the planar one. Both phases are separated by sparsely distributed Au atoms, which stabilize the flatness of the planar phase [58] and are rotated in respect to each other by 22° [59]. The determined rotation between the monolayers of both silicene phases is consistent with literature reports [63,64].

### 3.3. Destruction of the Silicene

#### 3.3.1. Thermal Modification of the Silicene—Domain Multiplication Stage

The low buckled and planar allotropes are generally thermally resistive and coexist in a wide temperature range. However, above 500 K, silicene undergoes smooth thermal modifications: clear changes in the morphology of the sample surface are observed. Roughly, three effects in diffraction patterns occur. One is that the spots of the low buckled phase of silicene become gradually blurry, and their intensity is slowly reduced, Figure 4a,b. The second is that the internal ring corresponding to the planar phase undergoes complex modification upon increasing temperature. The intensity of the diffraction spots of other domains of the planar silicene, which are barely visible at about 475–490 K, was comparable with the intensity of the main 24 spots (4 major domains), Figure 4c,d. The region outlined in Figure 4a,c is shown in Figure 4b,d after annealing at 475 K and 500 K, respectively. Thus, the formation of new rotational domains was observed, and consequently, the morphology of the planar silicene became more complicated.

The third effect is related to the superstructure diffraction spots connected with the twisted layers of both silicene phases, which are marked by the yellow, cyan and black arrows in Figure 4a. When the temperature approaches 500 K, the intensity of the diffraction spots of (√3 × √3), (√7 × √7) and (√21 × √21) superstructures significantly decrease, Figure 4c. Such an effect indicates that the order of the surface is gradually reduced. 

Further annealing above 530 K causes significant changes in the Au layer morphology. Initially, the flat and homogenous surface of the sample transformed upon increasing the temperature into a continuous film, consisting of regions of different thicknesses. Figure 5a–c presents LEEM images of the same surface area recorded with different electron energies.

Strong changes in the intensity of reflected electrons upon electron energy indicate the quantum size effect (QSE) occurring in regions of different thicknesses. Examples of reflectivity vs. electron energy curves taken at different sample regions are shown in Figure 4d. Various energy dependencies of the reflectivity curves clearly show different thicknesses of the considered sample regions. The observed huge changes in the film morphology did not destroy the planar silicene layer. It still existed on the surface, Figure 4e–j, although its visibility in the LEEM images was strongly influenced by the chosen energy of the electron beam. At energies corresponding to the minima of reflectivity, dark areas in Figure 4e–j, the hexagonal-like features were barely visible. 

At this stage of annealing, new rotated planar silicene domains usually appear. Moreover, as the temperature still increased, one can notice the formation of other domains with slightly larger lattices constant than estimated for the planar silicene 4.34 ± 0.05 Å, marked by ellipses in Figure 6.

Various lattice constants of the planar silicene were also reported in the previous STM experiments [58].

#### 3.3.2. Thermal Decomposition of the Silicene—Domain Destruction Stage

At still higher annealing temperatures, the observed changes of the surface morphology became even more dramatic. The Au layer cracked and consequently uncovered the silicon substrate. The de-wetting process was visible in both the LEED patterns and LEEM images. Figure 7 presents a series of LEED patterns taken during annealing of the Au layer from 540 K up to 565 K.

The obvious sign of the formation of separated Au islands was the appearance of the Si(1 × 1) together with the (√3 × √3) superstructure diffraction spots in the LEED pattern. They are marked in Figure 7 by small pink and yellow circles, respectively. They became sharp and intense with increasing annealing temperatures, indicating that more and more of the surface of the Si substrate became uncovered. It is important to note that, although very weak, there were still visible diffraction spots of the low buckled phase of silicene. On the other hand, the planar silicene spots were clearly visible, but their fingerprint no longer resembled what is known as the 24-spotted pattern. At the temperature of 540 K, one can distinguish only 18 main spots (3 domains) of the planar phase of silicene, marked by white, orange and green circles in Figure 7a.

Such a modified diffraction pattern shows that one of the main domains of the planar silicene disappeared at this temperature. Further temperature increase causes a gradual disappearance of the diffraction spots related to the remaining domains of the planar silicene. As a result, only 12 diffraction spots (two domains) and 6 diffraction spots (one domain) remained at 550 K and 560 K, respectively. Finally, at 565 K, residuals of silicene completely vanished, and only Si(1 × 1) together with the (√3 × √3) reconstruction remained. Numerous and very weak other diffraction spots were related to the (6 × 6)Au superstructures recovering after the Au de-wetting process.

The morphology of the layer after annealing at high temperatures depended on the Au film thickness. Thicker films revealed a much larger diversity of thicknesses of the Au islands, while the thinner ones are less rough. Figure 8 presents LEEM images of the transition stage from the continuous thick film to the appearance of the substrate registered for the 8 ML thick Au film.

Initially, the polygon-shaped features of the planar phase of silicene uniformly covered the whole surface. With increasing temperature, clear changes of surface topography took place: the area occupied by the silicene became reduced. The well-resolved planar silicene mesh survived in the form of small regions (one of such regions is marked by green arrow), Figure 8c–f. Simultaneously, Au-based droplets were formed, which grew with increasing temperature. In consequence, the Si substrate opened, red arrows in Figure 8e,f and only two regions at the sample surface can be distinguished: Si and Au droplets. 

Analysis of the intensity of the reflected electrons as a function of the energy of incident beam recorded for selected regions of the surface, Figure 8h, indicate similarity between the grey, shapeless areas and the droplets (pink, blue and green curves). They are clearly different from the red curve, which is taken from the bright regions of Figure 8g. This leads to the conclusion that, at this stage of annealing, the sample consisted of Au islands/droplets and the de-wetted Si(111) surface with (√3 × √3) or (6 × 6)Au superstructures. 

It is worthwhile to emphasize that the de-wetting process did not proceed homogeneously and was very sensitive to local conditions like substrate morphology, Au layer thickness and temperature. Therefore, various regions of the sample can be found at different stages of the transformation at the same time of thermal treatment. While a clean Si crystal surface decorated with Au droplets exists at a given location, residual silicene may still be present in its close vicinity, Figure 8i,j, respectively.

## 4. Conclusions

In conclusion, the thermally induced changes of the crystallographic structure and surface morphology of ultrathin Au layers grown at RT on the Si(111) surface were investigated by means of the low energy electron microscopy and low energy electron diffraction techniques. The diffusion of silicon atoms from the Si substrate through the Au film to its surface caused the formation of the low buckled phase of silicene. Although at room temperature the silicene layer was poorly ordered, the post-deposition sample annealing increased the size of silicene domains. The silicene layer was built of four rotated domains and remained as the only phase up to about 425 K. Above that temperature, besides the low buckled phase, the planar phase of silicene was also formed. Similarly, it consisted of four main domains. Each domain of the planar silicene had its counterpart in the low buckled phase. The domains of both phases were rotated in respect to each other, resulting in the appearance of characteristic superstructures in the LEED patterns: (√3 × √3), (√7 × √7) and (√21 × √21). Both silicene phases formed very well-ordered layered heterostructures covering the entire sample surface with the size of the order of cm^2^. Annealing at higher temperatures caused significant changes in the morphology of the Au layer. It formed large atomically flat regions of different heights, revealing a quantum size effect. At this stage, the Au layer remained continuous, and its top was still covered with the heterostructure of both phases of silicene. However, the characteristic superstructures became weaker, indicating a decreasing order of both silicene phases. A further temperature increase broke the Au layer. Finally, Au atoms formed droplets decorating the surface of the Si crystal where no silicene features existed anymore. Besides the Au droplets, the Si surface was covered with a very thin layer of gold atoms forming (√3 × √3) reconstruction. The present study allows for a better understanding of changes occurring on the surface of the Au/Si system upon its annealing between room temperature and the temperature at which the formation of Au droplets occurs. 

## Figures and Tables

**Figure 1 materials-15-01610-f001:**
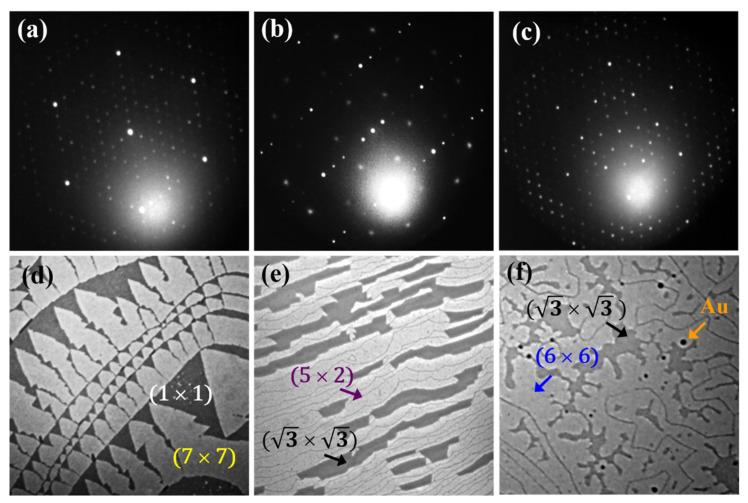
LEED patterns and corresponding LEEM images of: Si(111)-(7 × 7) surface (**a**,**d**), Si(111)-(5 × 2) coexisting with (√3 × √3) (**b**,**e**) and Si(111)-(6 × 6) coexisting with (√3 × √3) (**c**,**f**). LEED patterns were taken with *E* = 45.7 eV, whereas LEEM images (5 × 5 μm^2^) were recorded with *E* = 0.2 eV (**d**), *E* = 7.2 eV (**e**), *E* = 1 eV (**f**), respectively.

**Figure 2 materials-15-01610-f002:**
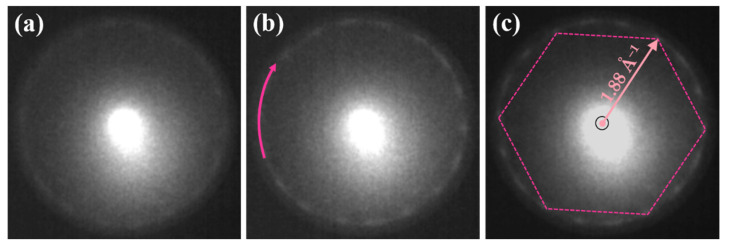
LEED patterns showing the thermal evolution of low buckled silicene recorded at RT (**a**), 410 K (**b**) and 420 K (**c**). The pink arrow in (**b**) indicates a ring with the diffraction spots of the low buckled silicene. The pink hexagon in (**c**) connects the spots of one of four domains. The black circle denotes the position of the (00) spot. All patterns collected at *E* = 14.2 eV.

**Figure 3 materials-15-01610-f003:**
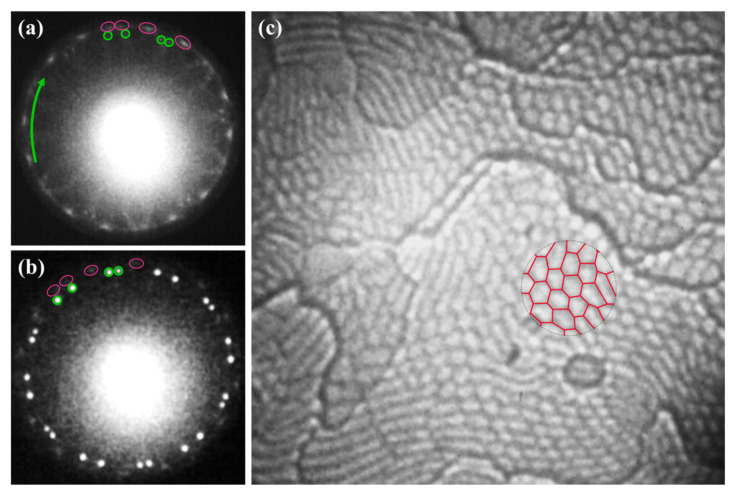
LEED patterns showing thermal development of low buckled and planar silicene recorded at 435 K (**a**) and 465 K (**b**). Green arrow in (**a**) indicates a ring with an additional set of diffraction spots representing planar silicene. Diffraction spots of four domains of both phases of silicene are marked by pink loops (low buckled) and green circles (planar). (**c**) Corresponding LEEM image with polygon-shaped features of planar silicene mesh recorded at RT, 1 × 1 μm^2^. In the central part, a red mesh showing polygon edge domains is superimposed. LEED patterns and LEEM images collected at *E* = 14.2 eV.

**Figure 4 materials-15-01610-f004:**
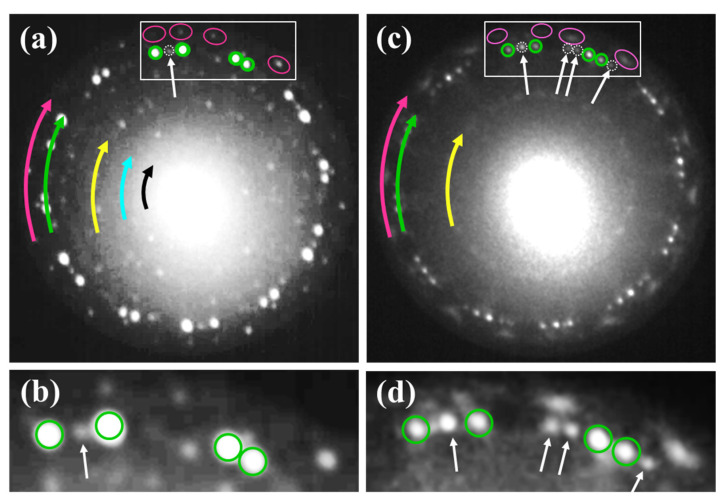
LEED pattern with fingerprint of well-ordered planar phase of silicene (ring marked by green arrow), recorded at 475 K (**a**). Twenty-four spots with strong intensity correspond to 4 major domains—four are marked by green circles. Barely visible additional domain is marked by white dashed circle. Almost invisible outer ring consisting of diffraction spots related to low buckled silicene is marked by pink arrow. Four selected spots representing four domains of low buckled silicene are marked by pink loops. Diffraction spots forming three inner rings correspond to (√3 × √3) (yellow), (√7 × √7) (cyan) and (√21 × √21) (black) superstructures. (**c**) LEED pattern of silicene recorded at 500 K. Additional diffraction spots indicated by white arrows appear. Enlarged region of diffraction pattern registered at 475 K (**b**) and 500 K (**d**). LEED patterns taken at *E* = 14.2 eV.

**Figure 5 materials-15-01610-f005:**
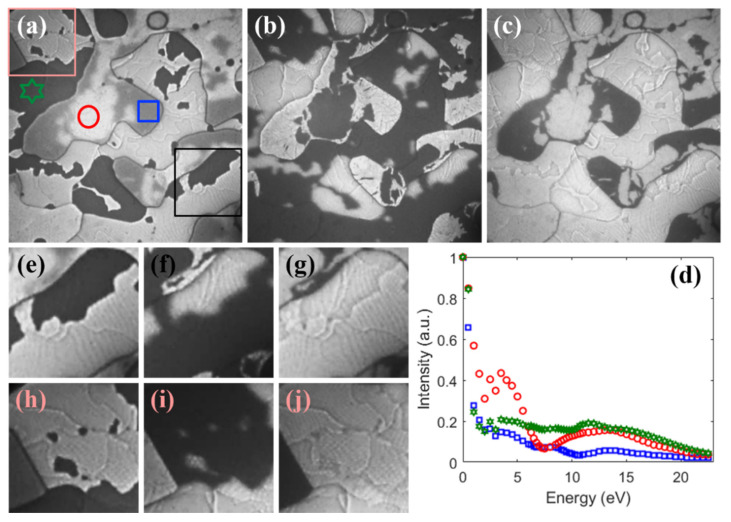
LEEM images of silicene annealed to 530 K and recorded at RT. Small squares in (**a**) denote regions where reflectivity curves (**d**) were measured. Large squares in (**a**) are magnified in (**e**–**j**). Images registered at: *E* = 3.7 eV (**a**), *E* = 6.7 eV (**b**), *E* = 9.9 eV (**c**), (**a**–**c**) 3.5 × 3.5 μm^2^, (**e**–**j**) 1 × 1 μm^2^.Color code and symbols of reflectivity curves (**d**) corresponds to color code and symbols of ROI in (**a**).

**Figure 6 materials-15-01610-f006:**
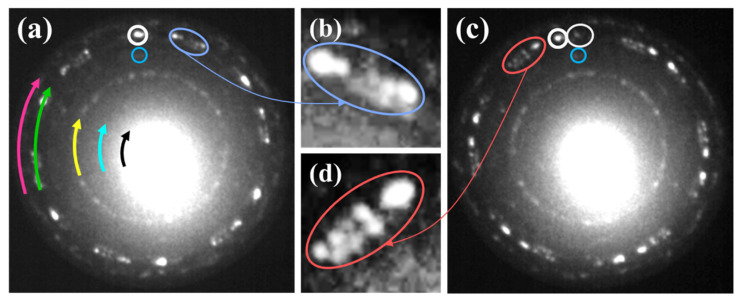
LEED patterns of silicene (pink arrow—low buckled phase, green arrow—planar phase) recorded at 530 K (**a**) and 535 K (**c**), *E* = 14.2 eV. Additional diffraction spots related to numerous domains of silicene with larger lattice constant visible in the enlarged regions of patterns (**b**,**d**—violet and purple ellipse). Position of selected single diffraction spot (white circle) and the groups of other diffraction spots (violet, purple and white ellipse) in relation to fixed position of barely visible spots of (6 × 6)Au superstructure marked by blue circle show dynamics of the system (**a**,**c**). Relatively intense Si(111)−(√3 × √3) periodicity visible as ring is marked by yellow arrow (**a**). The diffraction spots of the other two periodicities: (√7 × √7) and (√21 × √21), were very weak, cyan and black arrows, respectively.

**Figure 7 materials-15-01610-f007:**
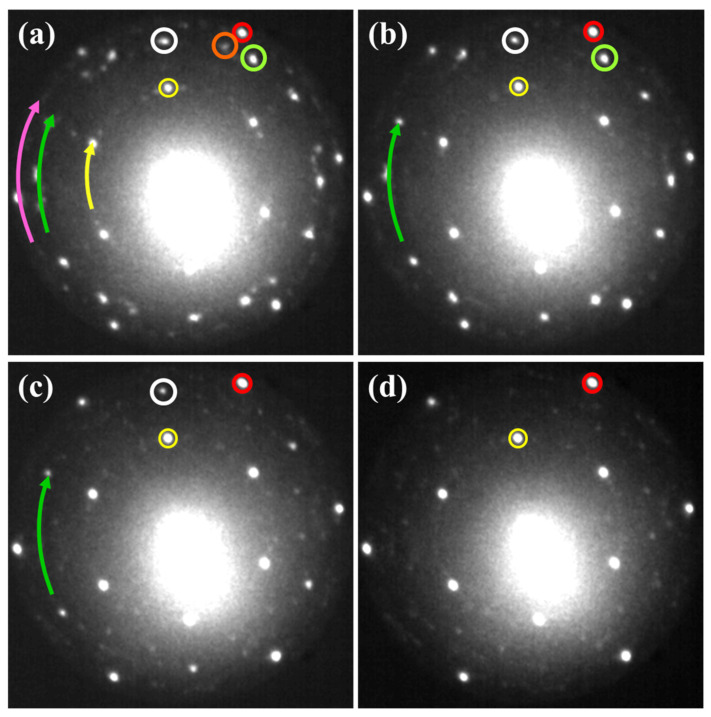
LEED patterns illustrating a process of degradation of low buckled (pink arrow) and planar (green arrow) phases of silicene: 540 K (**a**), 550 K (**b**), 560 K (**c**), 565 K (**d**). Twenty-four main diffraction spots of planar silicene as a consequence of destructive annealing are reduced to 18 spots (3 domains marked by white, orange and green circle (**a**)), and further to 12 diffraction spots (2 domains (**b**—white and green circle)) and 6 diffraction spots (1 domain (**c**—white circle)). Simultaneous decrease of intensity of (√3 × √3) periodicity denoted to mutual rotation of both silicene phases and in the same time increase of sharpness and intensity of the (1 × 1) and (√3 × √3) diffraction spots of Si substrate was observed, yellow arrow, red and yellow circles, respectively. At 565 K, mainly substrate with (1 × 1) and (√3 × √3) spots remained (**d**). All patterns were recorded at *E* = 14.2 eV.

**Figure 8 materials-15-01610-f008:**
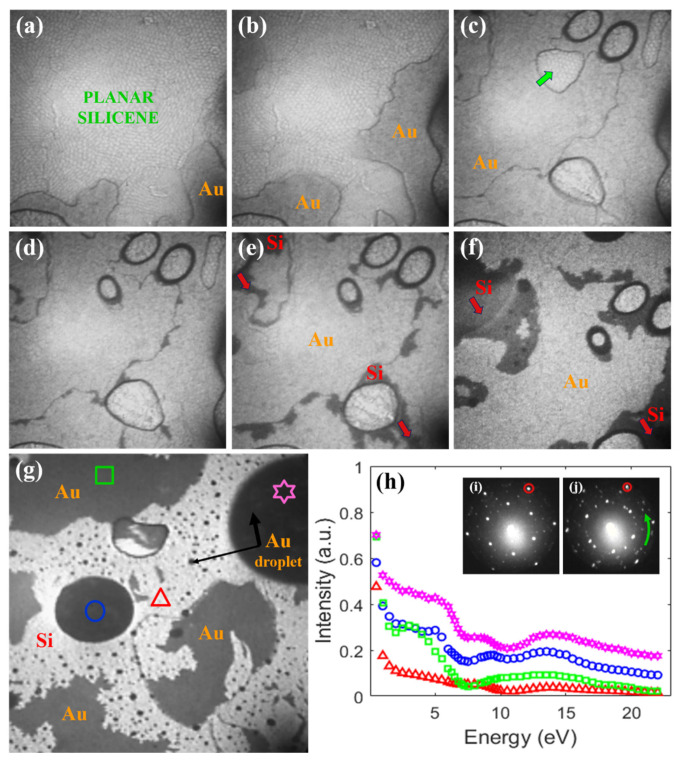
LEEM images of final stage of silicene overheating. Spilling Au reduces area originally occupied by planar silicene (**a**,**b**) to smaller regions (**c**–**f**). One of such regions is marked by a green arrow (**c**). De-wetting process uncovers Si surface (marked by red arrows ((**e**,**f**)—dark regions)) by Au cohesion into droplets. (**g**) Advanced stage of surface de-wetting with Au droplets marked by black arrows and opened Si substrate visible as bright areas. LEEM images recorded at: 540 K (**a**), 541 K (**b**), 542 K (**c**), 543 K (**d**), 545 K (**e**), 555 K (**f**), 565 K (**g**), with *E* = 11.2 eV (**a**–**f**) and *E* = 6.7 eV (**g**). All LEEM images: 3 × 3 μm^2^. LEED patterns recorded at *E* = 14.2 eV for different sample regions after annealing at 565 K with uncovered Si surface (**i**) and with silicene residuals (**j**). Intensity of reflected beam vs. energy of incident electrons (**h**) determined for regions marked by symbols in (**g**). Color code and symbols of reflectivity curves (**h**) correspond to color code and symbols of selected ROI (**g**).

## Data Availability

Data is contained within the article.

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
