# Peer review of "Temperature-Dependent Growth and Evolution of Silicene on Au Ultrathin Films—LEEM and LEED Studies"

_materials, 2022, doi:10.3390/ma15041610_

Round 1

Reviewer 1 Report

The manuscript titled "Temperature dependent growth and evolution of silicene on Au ultrathin films – LEEM and LEED studies" is worth for publication, So my recommendation is "as it as".

Author Response

We would like to thank the referee for his/her effort in positive evaluation of our manuscript.

Reviewer 2 Report

The present manuscript investigated the temperature-dependent growth and evolution of silicene in Au ultrathin films. The studies were performed using low energy electron microscopy (LEEM) and low energy electron diffraction (LEED). In general, the work is well written and detailed. Therefore, the manuscript may be for publication in its current form.

Author Response

(The authors gave the same response as above.)

Reviewer 3 Report

Review report

I have found the manuscript "Temperature dependent growth and evolution of silicene 2 on Au ultrathin films – LEEM and LEED studies" very well written. Authors have clearly presented their findings of fabricating silicene on Au thin films by adjusting the annealing temperature. I think this manuscript can be accepted with minor changes as detailed below:

  • The authors have mentioned that “…The pink hexagon in Figure 2(c) connects spots of one of the domains. From the distance between those spots and the (00) spot the lattice constant of 3.85 Å ± 0.05 Å has been obtained….”
    • Figure 2, please indicate (00) spot and the reciprocal distance between the spots on the pink hexagon and the (00) spot that was used to measure the lattice constant. Moreover, show for the readers, how the lattice constant was calculated.
  • The caption of Figure 3 mentions “….(c) Corresponding LEEM image with hexagonal-like features of planar silicene mesh, 1  1 m2. LEED patterns collected at E = 14.2 eV.”,
    • please indicate on the LEEM image which hexagonal features you re referring to, you can indicate them with arrows. And superimpose hexagonal model on the LEEM image if you like.
  • Figure 4, you need to show that different rings correspond to (√3´√3), (√7´√7), and (√21´√21).

Author Response

We would like to thank the referee for his/her effort in evaluation of our manuscript. In the following we shall answer all the points raised by the Referee.

Reviewer 4 Report

The manuscript presents a detailed study of the thermal evolution of the silicene layer obtained on Au film on Si(111) by LEED and LEEM. The study is complete and quite well discussed and presented, with minor revisions to be considered. The work was already partially described in a previous publication [59], even if here it is presented in a bit more extended way.

Minor points to be considered:

  1. Which is the Au thickness of the main part of the results presented? In the final part it is evidenced that Au thickness determines the behavior of the silicene film in temperature, but for the first part the Au thickness is not mentioned.
  2. No atomic model of the two structures or of the domains are reported in the manuscript. I understand this would increase the number of figures and the length of the manuscript (that is long enough) and that they are already discussed in a previous work. However, please consider including some model especially in the main LEED figures to help follow the discussion.
  3. Fig 3: What is the temperature of the LEEM image? 465 K? Please specify. Are only the planar regions visible? Are there also the low buckled areas? If yes please identify them.
  4. Fig 4: It would be  maybe clearer if the spots of low buckled phase are highlighted also in panel a. A mistyping is present in caption on the last line, d and e lettering of panels is wrong.
  5. Beginning of page 8: can authors give a tentative explanation for the presence of more lattice parameters in planar silicene?
  6. In section 3.3.2 it is described the decomposition of silicene and the appearance of Si substrate. In the discussion the appearance of the sqrt(3)xsqrt(3) is associated to this decomposition, while in the first section the same spots where assigned to a superstructure due to superposition of the two phases of silicene. Can authors explain better this point? Is the superstructure due to underlying Au or due ti silicene? Are they coincident?
  7. Fig. 7: and more in general, it would be useful to the reader that the color code is maintained throughout the manuscript. In panel a, pink is used for the Si substrate spots, while previously it was used for low buckled silicene. It would be better to use a different color (same for inset of Fig. 8h). For the same reason colors and arrows should be reorganized in Fig. 8, where variations of blue are used to indicate features in LEEM images, but they are hard to recognize, while Au tags are not corresponding to arrows (of different color). This complicates the understanding of the features in the images.

In general it would be appreciated a short description of the structures associated to the observed complex LEED patterns and to the different phases in the film at the various temperatures.

Author Response

(The authors gave the same response as above.)

Reviewer 5 Report

This article reports a detailed analysis performed by LEEM and LEED on 2D silicene growth layer which was formed through Si diffusion from Si (111) substrate via gold ultrathin layer to the top of gold surface. Silicene growth increases with increasing the annealing temperature. Three different stages on domain phase changes were investigated by LEEM and LEED based on surface morphology and diffraction patterns of the silicone annealed between room temperature and temperature at the formation of Au droplets on Si (111) surface. It is a good analytical article to provide a better basic understanding of structure and morphology changes at the Au/Si interface.  Questions:

  1. How do you deposit Au ultrathin film by what processing method? Need to briefly mention in the Materials and Methods section. Not clear.
  2. What is the process method to control and measure the monolayer deposition of Au ultrathin film? Need to briefly mention in the Materials and Methods section. Not clear.
  3. In Fig. 4, the label (d) should be (b) and the label (e) should be (d)
  4. In line 26, one element of group “13 ¸ 16” of the periodic table, it should be …group “13 – 16” of.., a typo.
  5. In line 71, a specific resistivity of “3 ¸ 8” Wcm at room temperature (RT) should be ...of “3 – 8” W., a typo.
  6. Some minor English gramma errors in the text need to be edited by a native English speaking person or a professional editor in English, if possible.

Author Response

(The authors gave the same response as above.)
